# Acute and 28-Day Repeated-Dose Oral Toxicity of the Herbal Formula Guixiong Yimu San in Mice and Sprague–Dawley Rats

**DOI:** 10.3390/vetsci10100615

**Published:** 2023-10-10

**Authors:** Ling Wang, Jiongjie He, Lianghong Wu, Xueqin Wu, Baocheng Hao, Shengyi Wang, Dongan Cui

**Affiliations:** Key Laboratory of New Animal Drug Project, Gansu Province; Key Laboratory of Veterinary Pharmaceutical Development, Ministry of Agriculture and Rural Affairs, Lanzhou Institute of Husbandry and Pharmaceutical Sciences of Chinese Academy of Agriculture Sciences, Lanzhou 730050, China; wangling02@caas.cn (L.W.); ajie@163.com (J.H.); 17361637229@163.com (L.W.); haobaocheng@caas.cn (B.H.)

**Keywords:** herbal remedy, Guixiong Yimu San, safety assessment, maximum-tolerated dose (MTD), oral toxicity

## Abstract

**Simple Summary:**

Postpartum uterine diseases of dairy cows include retained placenta (RP) and puerperal metritis (PM). Cows suffering from a retained placenta are at an increased danger of developing puerperal metritis or endometritis, and subsequent decreased fertility. The dominant approach to a retained placenta in cows is the local or systemic administration of antibiotics; however, current reports point to the low efficacy of intrauterine and systemic antibiotics in accelerating the separation and removal of the retained placenta. Therefore, substitute approaches to cope with these two disorders in postpartum cows should be developed. Herbal medicines are considered good alternatives to allopathic medicine, and some ethno-veterinary medicines are often used to treat the RP in cows. GYS is primarily used to treat postpartum blood stasis syndrome caused by retained placenta. It was designed based on the therapeutic role of enhancing the blood flow and removing blood stasis, maintaining the qi movement, and releasingthe pain.It can create a very promising state for prior placental detachment and the spontaneous discharge of the RP, but its toxicity has not been completely assessed. In the present study, we demonstrated that GYS did not have any therapeutic side effects in rats, thus providing a practical approach for the clinical control of a retained placenta.

**Abstract:**

To evaluate the acute and chronic 28-day repeated-dose oral toxicity of Guixiong Yimu San (GYS) in mice and rats, high-performance liquid chromatography (HPLC) was used to determine the stachydrine hydrochloride in GYS as the quality control. In the acute toxicity trial, the mice were administered orally at a dose rate of 30.0 g GYS/kg body weight (BW) three times a day. The general behavior, side effects, and death rate were noticed for 14 days following treatment. In the subacute toxicity trial, the rats were administered orally at a dose rates of30.0, 15.0, and 7.5 g GYS/kg BW once a day for 28 days. The rats were monitored every day for clinical signs and deaths; changes in body weight and relative organ weights (ROW) were recorded every week, hematological, biochemical, and pathological parameters were also examined at the end of treatment. The results showed that the level of stachydrine hydrochloride in GYS was 2.272 mg/g. In the acute toxicity trial, the maximum-tolerated dose of GYS was more than 90.0 g/kg BW, and no adverse effects or mortalities were noticed during the 14 days in the mice. At the given dose, there were no death or toxicity signs all through the 28-day subacute toxicity trial.The oral administration of GYS at a dose rate of 30.0 g/kg/day BW had no substantial effects on BW, ROW, blood hematology, gross pathology, histopathology, and biochemistry (except glucose), so 30.0 g/kg BW/day was determined as the no-observed-adverse-effect dosage.

## 1. Introduction

A retained placenta (RP) is a frequent uterine disorder in first postpartum cattle [1,2]. Cows with an RP have an increased danger of developing puerperal metritis, and the consequences of an RP on farm productivity partially depend on the occurrence of metritis [3,4,5]. Thus, the medical therapy for a retained placenta is an ideal means to avoid puerperal metritis in cows. Usually, the principal strategy to an RP in cattle is the local or systemic administration of antibiotics [6,7]. Previous studies [6,7] have demonstrated the benefits of an intrauterine antibiotic treatment for the prevention of metritis in cows with an RP. Ceftiofur, a 3rd generation cephalosporin drug does not leave antibiotic residues in the milk when used in line with the label guidelines and has therefore been useful in clinical practice for treating cows with a retained placenta and preventing metritis [8,9,10]. However, current reports point to the low efficacy of intrauterine and systemic antibioticsin accelerating the separation and removal of the RP [11,12,13]. Therefore, substitute approaches to cope with these two disorders in postpartum cows should be developed. Traditional Chinese veterinary medicine (TCVM) has always been more focused on the body’s reaction to pathogenetic causes compared to pathological pathways [6,14,15], and thus, it may offer novel possibilities for the management of post-delivery uterine disorders in cows.

Guixiong Yimu San (GYS) has been used for the clinical treatment of a retained placenta with satisfactory therapeutic effects (the therapeutic oral dose for cows is approximately 0.5 g/kg BWonce per day for 1–3 days), and it can also lower the risk of puerperal metritis in postpartum cows [16,17]. GYS used in the present study consisted ofsix Chinese herbs, including Leonuri herba (Yi-Mu-Cao, YMC; *Leonurus japonicus* Houtt.), Angelicae sinensis radix (Dang-Gui, DG; *Angelica sinensis* (Oliv.) Diels), Chuanxiong rhizoma (Chuan-Xiong, CX; *Ligusticum chuanxiong* Hort.), Carthami flos (Hong-Hua, HH; *Carthamus tinctorius* L.), Cyperi rhizoma (Xiang-Fu, XF; *Cyperus rotundus* L.), and Glycyrrhizae radix et. rhizoma (Gan-Cao, GC; *Glycyrrhiza uralensis* Fisch.). It was designed based on the therapeutic role of enhancing blood flow and removing blood stasis, maintaining the qi movement, and releasing the pain. All of the herbal plants work collectively as a recipe to increase the blood flow, convert stasis, contract of the uterus, warm up the channels, and relievethe pain to resolve the postpartum blood stasis syndrome. By removing blood stasis from the retained placenta, regulating the uterine contractions, and improving the uterine state, GYS can create a much more promising situation for earlier placental separation and the spontaneous release of the RP [18,19,20]. Numerous phytochemical compounds, such as alkaloids, saponins, flavonoids, volatile oil, ferulic acid, and others, have been identified in the herbs contained in GYS [21,22,23,24]. In accordance with the composition of the traditional Chinese medicine formula, the preferred principle of Monarch drug (JUNYAO), and the instruction of the Committee for the Veterinary Pharmacopoeia of China (2015 version) [25], stachydrine hydrochloride, the biologically active chemical compound of Leonuri herba (Yi-Mu-Cao, Monarch drug, the key component of herbal formula), was indicated to be a biomarker of GYS for qualitative identification and quantitative analysis, which may improve the level of quality control and promote the internationalization of TCMs.

Pharmacological studies and clinical trials of GYS have been performed, and multiple scientific researchers have confirmed its therapeutic value, thereby attracting growing attention to the application of GYS [17,20]. Although, to the best of our information, there has been no systemic research on the subacute toxicity of GYS to support clinical drug safety, and generally very little is known about the toxicity of GYS. Along with the development of a potent herbal product, the evaluation of adverse effects on living organisms is one of the key issues; therefore, further studies should establish the toxicity profile of GYS by evaluating its safety parameters using the standardized trial protocols from regulatory organizations. In the present study, the safety of GYS was evaluated through oral administration by undergoing the acute and subacute toxicity trials in mice and rats, respectively. The acute and 28-day repeated-dose oral toxicity experiments of GYS are essential for clarifying the drug’s toxicity, and providing a basis for the subsequent application of GYS. Moreover, such experiments may assist in treating the cows with an RP and the prevention of PM under field circumstances, and may provide guidance for phase II clinical trials.

## 2. Materials and Methods

### 2.1. Drugs and Reagents

Guixiong Yimu San (GYS, batch number: 20210618) was purchased from Beijing Kangmu Biotechnology Co., Ltd. (Beijing, China). The GYS consisted of six herbal materials, i.e., Leonuri herba (Yi-Mu-Cao, YMC; 46.15%, *w*/*w*; JUNYAO, Monarch drug—the key component of herbal formula), Angelicae sinensis radix (Dang-Gui, DG; 15.38%, *w*/*w*), Chuanxiong rhizoma (Chuan-Xiong, CX; 13.46%, *w*/*w*), Carthami flos (Hong-Hua, HH; 9.62%, *w*/*w*), Cyperi rhizoma (Xiang-Fu, XF; 7.69%, *w*/*w*), and Glycyrrhizae radix et rhizoma (Gan-Cao, GC; 7.69%, *w*/*w*). The final concentration of the GYS solution was 3.0 g/mL (*w*/*v*), and it was stored at −20 °C before administration.

For HPLC analysis, stachydrine hydrochloride—the active compound of Leonuri herba (Yi-Mu-Cao)—was chosen as the biomarker in the HPLC-based quality control. Stachydrine hydrochloride was obtained from the National Institutes for Food and Drug Control (Purity > 98%, Batch No.: 110712-201614, Beijing, China). Stock solution of the relevant criteria was made at the appropriate concentration (0.0375 mg/mL, *w*/*v*) in methanol and kept at 4 °C up tofurther analysis. Ethanol and acetic acid were obtained from Shanghai Chemical Reagent Company, China. Acetonitrile and methanol (HPLC grade) were obtained from ThermoFisher Scientific (Waltham, MA, USA).

### 2.2. HPLC Analysis

The content of stachydrine hydrochloride—the active compound of *Leonuri herba* (Yi-Mu-Cao; JUNYAO, Monarch drug), was determined in GYS by HPLC (Chinese Veterinary PharmacopoeiaCommission, 2015; Chinese Pharmacopoeia Commission, 2020) [25,26]. Quantitative analysis of GYS was performed on Agilent 1290 Infinity G4218A, Sedex-85, Evaporative Light Scattering Detector (ELSD) (Agilent, Santa Clara, CA, USA). The column was Venusil HILIC chromatographic column (4.6 mm × 250 mm, 5.0 µm; Agela Technology Co., Ltd., Tianjin, China). The mobile phase comprised a cetonitrile (A) and 0.2% (*v*/*v*) acetic acid aqueous solution (B) (87:13), and the pH value ranged from 6.8 to 7.2. Chemicals were sieved through a Millipore 0.45 mm filter and degassed before use. The conditions were as follows: volume flow rate, 1.0 mL/min; evaporative light scatterer; column temperature, 25 °C; the temperature of the detector, 50 °C; the temperature of the atomizing chamber, 80 °C; and the injection volume, 10 µL. The theoretical plate number was not less than 5000 according to the peak of stachydrine hydrochloride. Data assembly and quantification were performed with Agilent ChemStation and Agilent EZChrom Elite (Agilent, Santa Clara, CA, USA). The peak of the stachydrine hydrochloride was recognized by comparing with standard chemicals.

### 2.3. Experimental Animals

Thirty Kunming (KM) mice (18–22 g, male:female = 1:1) were used for the 14-day acute toxicity trial. A total of 80 Sprague–Dawley (SD) rats, age 7–8 weeks (160–180 g, 40 males and 40 females) were used for the 28-day repeated-dose oral toxicity trial. The animals with a clean grade (Certificate No.: SCXK (G) 2020–0002) were received from the Laboratory Animal Center of Lanzhou Veterinary Research Institute (Lanzhou, China). All of the experimental animals were kept in plastic cages and kept separately according to sex under the following conditions: temperature, 22 ± 2 °C; relative humidity, 55 ± 10%; and a 12 h light/dark cycle. They were fed a standard pelleted diet (Test clean grade, granule rat feed, Nanjing, China) and ad libitum water access. The animals underwent 7-day acclimatization before the experimentalprocedures.

All of the experimental trials performed in the present study were conducted in accordance with the guidelines of the National Institutes of Health (NIH, 2015) [27] for the Care and Use of Laboratory Animals. The animal study was reviewed and approved by the Ethics Committee of the Lanzhou Institute of Husbandry and Pharmaceutical Sciences of the Chinese Academy of Agricultural Sciences (Lanzhou, China).

### 2.4. Oral Acute Toxicity of GYS in Mice

The acute oral toxicity of GYS was assessed in the mice by the maximum-tolerated dose (MTD) method, in accordance with the method of the Compilation of Technical Guidelines for Veterinary Drug Research (Center for Veterinary Drug Evaluation (CVDE), Ministry of Agriculture of the People’s Republic of China, 2015) [28], and Organization for Economic Cooperation and Development (OECD) Guideline No. 423 (OECD, 2002) [29]. Fifteen male and female mice each go through a 7-day isolation and adaptation process. Following overnight fasting, 30.0 g GYS/kg body weight (BW) was administered orally through an oral gavage needle three times a day (total of 90.0 g GYS/kg BW/day), with a 6 h of interval among the doses. The common activities of the mice were noted down after every 4 h of treatment and 24 h for fourteen days. Towards the end of the trial, the animals were euthanized, and gross pathological changes in the vital body organ such as liver, heart, lung, spleen, gastrointestinal tract, kidneys, thymus, testes, and ovaries, were examined. Histopathological analyses were performed by H&E staining if the gross pathological changes were recorded in vital organs [30,31].

### 2.5. Oral 28-Day Repeated Toxicity Trial of GYS in Rats

The subacute toxicity trial was carried out in agreement with the Rules of Animal Drug 28-Day Repeated Dose Toxicity Study (CVDE, Ministry of Agriculture of People’s Republic of China, 2015) [28], and followed the protocol described by OECD Guideline No. 407 (2008) [32]. Eighty SD rats were randomized into a control group (CG) and three treatment groups (n = 20; 10 male and 10 female). GYS was administered once a day by oral gavage for 28 continuous days at three different doses, i.e., at 30.0, 15.0, and 7.5 g GYS/kg BW, respectively (at 8:00–9:00 a.m.), leading to the distinction of three GYS treatment groups; namely, the high-dose group (30.0 g GYS/kg BW/day), the medium-dose group (15.0 g GYS/kg BW/day), and the low-dose group (7.5 g GYS/kg BW/day). The control group received only physiological saline (0.9% normal saline). All of the experimental animals were monitored for behavioral activities, deaths, and signs and symptoms of gross toxicity twice a day through the 28-day subacute toxicity trial. At the end of the drug administration period (24 h after the last administration), all surviving rats from each group were euthanized for blood hematology, serum biochemistry, histopathology, and relative organ weight (ROW) [33,34].

### 2.6. Clinical Observations and BW Monitoring

Clinical observations were noted once a day to record thetoxicity signs, particularly at a similar time every day (1 h after GYS or vehicle administration). The routine observation focused on fluctuations in general performance, movement, fur, skin, and eyes with particular consideration given to noting the occurrence of lethargy, diarrhea, tremors, coma, convulsions, and sleep. The BW of each rat was noted before the administration of GYS at day 1, every 7th day thereafter, and necropsy day. The BW at necropsy was noted after 16 h fasting. The average body weight gain (BWG) of a week was measured for each sex and dosage level throughout the whole trial period (Appendix A). Additionally, the amounts of feed given and their leftovers on the following day were estimated to determine the difference which was considered as daily feed consumption (Appendix A). The average daily feed utilization was estimated for each dosage and sex for every week interval and the total trial period.

### 2.7. Hematological and Biochemical Analyses

At the end of the trial, all the surviving rats were fasted for 16 h (water allowed) before blood sampling and sedating with sodium pentobarbital (50.0 mg/kg BW, i.p.). Blood was collected via the cardiac rupture into EDTA-containing and non-heparinized vacutainer tubes for hematology and biochemistry parameter analyses, respectively. The following indicators such as white blood cell count (WBC), monocyte count (MONO#), lymphocyte count (LYM#), lymphocyte % (LYM%), monocyte % (MONO %), granulocyte count (GRAN#), granulocyte % (GRAN %), red blood cell count (RBC), red blood cell distribution width (RDW), platelet count (PLT#), mean platelet volume (MPV), platelet distribution width (PDW), hemoglobin (HGB), hematocrit (HCT), mean corpuscular volume (MCV), mean corpuscular hemoglobin (MCH), mean corpuscular hemoglobin concentration (MCHC), and platelet cubic measure distribution width (PCT), were measured using an automatic five-class animal blood cell analyzer (ProCyte Dx, IDEXX Laboratories Inc., Westbrook, ME, USA) by fixing the biological reference intervals for the sex and strain of rats.

Blood samples for serum biochemical assays were kept for 2 h to clot, and centrifugation was performed at 4000 rpm and 4 °C for 10 min and then stored at −20 °C until further analysis. The main parameters included were alkaline phosphatase (ALP), alanine aminotransferase (ALT), aspartate aminotransferase (AST), creatinine (CRE), glucose (GLU), total bilirubin (TBL), total protein (TP), blood urea nitrogen (BUN), total cholesterol (TC), triglyceride (TG), and albumin (ALB), and were measured with an automatic serum biochemistry analyzer (BS-420, Shenzhen Mindray Bio-Medical Electronics Co., Ltd. Shenzhen, China) by fixing the biological reference intervals for the sex and strain of the rats. Samples were protected from light by using aluminum foil, especially for bilirubin.

### 2.8. Gross Necropsy and Organ Weights

Necropsies were carefully conducted on all of the rats. Later to blood collection, the rats were euthanized by exsanguination from the aorta and vena cava. The animals were carefully observed for external defects. The thoracic and abdominal cavities were also observed for defects. Moreover, selected vital organs (heart, lungs, liver, spleen, and kidney), reproductive (uterus and ovary or testis), the stomach, and intestines were examined macroscopically. Then, the organs were quickly detached, balanced, and secured in 10% neutral buffered formalin solution (SolarBio Technology Co., Ltd., Beijing, China) for at least 48 h. The ROW was estimated using the following formula; organ weight/body weight on euthanize day × 100% (Appendix A).

### 2.9. Histopathological Analyses

After fixation, the selected tissues were dried up in a high-grade ethanolsolution with 70, 90, 95, and 100% and treated with ethyl salicylate. After embedding in paraffin and cutting to 5 µm-thick pieces, the tissue pieces were dyed with hematoxylin and eosin (H&E) for photo-microscopic analysis.

### 2.10. Statistical Analysis

Data were expressed as the mean and standard deviations (mean ± SD). Mean differences among the control and treatment groups were evaluated by one-way analysis of variance (ANOVA), followed by Tukey’smultiple comparison and Dunnett-t post-hoc test for 28-day repeated-dose toxicity (SPSS Statistics 19.0, IBM, Chicago, IL, USA). Graphs were drawn using the GraphPad Prism 5.0 (GraphPad Prism, San Diego, CA, USA). *p <* 0.05 was considered as significant.

## 3. Results

### 3.1. HPLCAnalyses

The HPLC chromatogram of GYS is displayed in Figure 1. As the active compound of Leonuri herba (*Leonurus japonicus* Houtt., JUNYAO, Monarch drug), stachydrine hydrochloride was found in GYS at 2.272 mg/g through HPLC analysis.

### 3.2. Acute Toxicity

During the 14-day acute toxicity test, neither mortality nor abnormalities in behavior (lethargy, sleep, coma, tremor, and diarrhea) were observed in any of the mice. In addition, no significant pathological changes in the colors and textures of vital organs, including the heart, lung, liver, kidneys, spleen, thymus, ovaries or testes, and gastrointestinal tract, were found by macroscopic examination. On that basis, the LD50 of GYS was estimated to be more than 90.0 g/kg BW by oral route.

### 3.3. Subacute Toxicity–BW and Feed Consumption Observations

No treatment-related deaths or behavioral changes occurred during the trial period. There were no abnormal changes in clinical signs, BW, or BWG caused by the oral administration of GYS in either sex, and there were no significant variations relative to the control group (*p* > 0.05). Feed consumption was slightly changed in the three GYS-treated groups with no significant statistical difference from the control group (*p* > 0.05), illustrating that GYS had no obvious effect on the average daily gain (ADG) and average daily food intake (ADFI). Furthermore, the male rats had more significant ADG and ADFI values than the female rats, and their stage (weekly) growth advantage was more significant (Figure 2, Table 1).

### 3.4. Subacute Toxicity–Hematologicaland Serum Biochemical Analyses

As revealed in Table 2, all the measured blood parameters were slightly increased or decreased, i.e., there was the reduction in WBC, LYM, GRAN, RBC, PLT, and PDW in the male and female rats, and a slight increase in MONO and MPV in both sexes, but there were no remarkable differences (*p* > 0.05) among the three GYS-treated groups and the control group (Table 2). These data showed no clear dose-response relationships and all of the minor fluctuations in the following parameters in either sex wereinside the normal range of the testing research laboratory. The raw data for hematology and biochemistry parameters in the treatment and control male and female rats is enclosed in Appendix A. Subacute oral administration of GYS for 28 daysdid not affect significant variations (*p* > 0.05) in most of the biochemistry parameters, comprising serum ALP, TBL, BUN, CRE, TC, TG, TP, ALB, and the level of the indicator enzymes of the liver (AST and ALT) in both the male and the female rats (Table 3). However, the levels of GLU in both sexes on day 28 with three different doses (7.5, 15.0, and 30.0 g GYS/kg BW) decreased significantlyrelative to the control group (*p* < 0.05).

### 3.5. Subacute Toxicity–ROW Analyses

After administration for 28 consecutive days, there were no statistically remarkable differences (*p* > 0.05) in the ROWs of the organs in the male and female groups at any dosage level. All of the slight variations in these ROW values were inside the normal range (Table 4).

### 3.6. Subacute Toxicity–Gross Necropsy and Histopathological Observation

The appearance and color of the vital and reproductive organs of the rats at necropsy were normal in all of the GYS-treated groups. There were no effusions or adhesions in the chest cavity, abdominal cavity, pericardium, or viscera, and no visible lesions such as congestion or bleeding, edema or swelling, or hypertrophy or atrophy. Microscopic observations did reveal any evident pathological modifications linked with the administration of GYS. The histopathological analysis involved the microphotographs of the small intestine, stomach, liver, heart, lung, spleen, kidney, testes, uterus, and ovaries of the rats in the three GYS-treated groups and the control group (Figure 3 and Figure 4). We found no histopathological changes in the organ sections of the rats after 28 days of GYS administration compared with the control group, illustrating that GYS had no obvious effect on the vital organs and reproductive organs and that GYS fed at 30.0 g/kg BW for 28 days was generally safe for rats. Thus, the no-observed-adverse-effect level (NOAEL) in the present study was 30.0 g GYS/kg BW.

## 4. Discussion

Recently, the area of complementary/alternative medicine has been taking on new prospects. Herbal medicines are considered good alternatives to allopathic medicine and are becoming common globally, in addition to being recommended, accompanied by Western treatments by several doctors [35,36]. Ethno-veterinary medicine plays a keypart in animal well-being globally [37,38,39]; e.g., some therapeutic plants are frequently used to treat cows with an RP [40,41,42]. Though, herbal remedies are generally not subject to toxicity trials prior to clinical uses in contrast with Western medical treatments. As many herbal drugs are being recommended, their safety has turned into a main issue. Therefore, herbal medicines also need a deeper assessment of their effectiveness and protection using standard toxicological techniques due to the need for testified medical use earlier to clinical uses [43].

The postpartum uterine diseases of dairy cows include a retained placenta and puerperal metritis. Cows with an RP are at an increased risk of puerperal metritis or endometritis, and consequent decreased fertility [2,4]. Evidence from previous studiesproposes the important effects of herbal medicines for postpartum uterine disorders in dairy cattle [17,20,44,45]. GYS is a traditional Chinese herbal formula that is principally used as a therapeutic drug for postpartum blood stasis syndrome caused by an RP, which is a frequently diagnosed uterine disorder in early postpartum cows. In the TCVM concept, blood inertia is a vital primary pathology of certain postpartum disorders [46,47]. Moreover, RP and/or PM both lie under the category of a blood stasis syndrome (Editorial Committee of Encyclopedia of China’s Agriculture, 1991) [48]. Although GYS has previously been revealed as an effective herbal formula in clinical practice, its toxicity has not been fully estimated. In the present study, we conducted an acute toxicity trial in KM mice and a subacute toxicity trial in SD rats to assess the safety level of GYS.

Additionally, the quality control of medicinal agents and formulation with recent analytical equipment is increasingly essential to ensure their effectiveness. Leonuri herba (Yi-Mu-Cao) is a fresh or dried ground part of *Leonurus japonicus* Houtt. From the *Labiaceae* family. Generally, it is used as a medicine for female reproductive diseases, such as a remedy to treat delayed menstruation, premenstrual tension, and menstrual pain, and it is formally listed in the Chinese Pharmacopoeia of PR China (Chinese Pharmacopoeia Commission, 2020) [26] and Chinese Veterinary Pharmacopoeia of PR China (Chinese Veterinary Pharmacopoeia Commission, 2015) [25]. According to modern pharmacology research, it also has emmenagogue, cardiotonic, and anticoagulant activities [20,23]. The largest component of GYS is Leonuri herba—Monarch drug (JUNYAO, the importantconstituent of herbal formula)—which can enhance the uterine contractions, improve the uterine hemorheology and microcirculation against the primary indication of cows with an RP, and provide the principal therapeutic effects against postpartum blood stasis syndrome [49]. In this study, the active compound (stachydrine hydrochloride) of Leonuri herba was identified and analyzed using HPLC. The study revealed the presence of this active compound in GYS (2.272 mg/g), suggesting that the quality of GYS is improved under the present situation. Leonuri herba (Yi-Mu-Cao, YMC; 46.15%, *w*/*w*; JUNYAO, Monarch drug) is the key component of Guixiong Yimu San (GYS). In accordance with the composition of traditional Chinese medicine formula, the preferred principle of Monarch drug (JUNYAO), and the instruction of the Committee for the Veterinary Pharmacopoeia of China (2015 version), stachydrine hydrochloride, the biologically active chemical compound of Leonuri herba (the key component of herbal formula)—was indicated to be a biomarker of GYS for qualitative identification and quantitative analysis. The content of stachydrine hydrochloride is an important index component to measure the quality of GYS. The quantitative study of stachydrine hydrochloride by the HPLC method can effectively control the quality of GYS [22,49].

Our previous study suggested approximately 0.5 g/kg BW once per day for 1–3 days as the therapeutic oral dose of GYS for cows. In the acute toxicity trial, we used 90.0 g GYS/kg BW, which is nearly 60 times greater than the dosage that has previously been used in the clinical practice, to assess the MTD in the mice. In the current subacute toxicity study, three different doses (30.0, 15.0, and 7.5 g GYS/kg BW)—representing high, medium, and low doses, respectively—which are approximately 20, 10, and 5 times greater than the dosage that has previously been used in the clinical practice, respectively, were used to examine the NOAEL in the rats. The results from the acute toxicity trial revealed no adverse effects in the mice administered 90.0 g GYS/kg BW orally. Additionally, there were no anomalies detected in the mice vital organs after euthanasia. These findings showed that MTD for the oral gavage of GYS in the mice is higher than 90.0 g/kg BW which is too high than what has already been used in clinical practice. As per the CVDE Guidelines of Veterinary Drug Acute Toxicity Study (Center for Veterinary Drug Evaluation, Ministry of Agriculture of the People’s Republic of China, 2015) [28], the oral administration of GYS at a dosage lower than 90.0 g/kg may be known as principally nontoxic or slightly toxic at the most. During the 28-day oral administration of GYS, none of the rats died; there were no noteworthy changes in the usual activities, posture, mucosa, or color in any of the GYS-treated groups. Furthermore, the BW and ROW of the rats’ even at the high dose of GYS were not considerably different from those in the control group. Generally, reductions in BWG and ROW are mostly monitored markers of toxicity after contact with chemicals and drugs [34,50]. These outcomes propose that the longtime use of GYS with a high dose of 30.0 g/kg BW did not cause weight loss and delayed growth in the rats. Additionally, findings from a previous study [17,20] showed that GYS had no obvious effects on clinical features, daily yielding, and hematological indexes in dairy cows receiving an oral dose of 0.67, 2.00, or 3.33 g/kg BW (once daily for three consecutive days), which is consistent with our results. Indication from the randomized précised clinical trial suggests that GYS is a clinically effective therapeutic agent for RP and the prevention of PM; thus, it might have much potential for the clinical control of RP in dairy cows.

The hematopoietic system is a delicate object for lethal substances, and variations in hematology parameters have high predictive significance for toxicity or pathological grade in humans using the data gained from the animal study model [51,52,53,54]. In the present study, at the last of the drug administration period, the standard hematological and biochemistry tests were performed to estimate the hematology parameters, metabolic products, enzymes, and substrates. The tested hematological parameters revealed no significant differences among the control and the three GYS-treated groups, representing that GYS had no effect on the circulating blood cells or their production. Moreover, the serum clinical biochemistry also plays a significant role in the analysis of toxicity prompted by various drugs and chemical materials. Biochemical parameters such as AST, ALT, ALP, TP, albumin, and TBL are used as good indicators of hepatic functions; AST and ALT are well-recognized liver enzymes which are used as biomarkers to calculate possible toxicity, while BUN and creatinine levels are good signs of renal function [55,56,57]. In the present study, there were no adverse effects on the common biomarkers of liver and kidney toxicity, so it can be summarized that GYS did not induce remarkable injury to these organs. Usually, any injury to the parenchymal liver cells results in elevated levels of both transaminases (AST and ALT) and ALP in the blood, and AST found in the serum is of both mitochondrial and cytoplasmic origin, so any increase can be taken as an initial sign of cellular damage that results in the outflow of these enzymes into the serum. In this study, no statistically significant differences (*p* > 0.05) were noted in the levels of ALT, AST, and ALP in male and female rats, indicating that the subacute administration of GYS did not change the hepatocytes’ function and metabolism. There was also no marked change in creatinine in the three GYS-treated groupscompared with the control group. Creatinine and blood urea nitrogen are considered valid indicators of renal function, and if there is marked damage to the functional nephrons, the increase in creatinine and BUN levels is expected. Thus, the findings from the present study imply that GYS did not affect kidney function. Moreover, no statistically significant differences (*p >* 0.05) were observed in the levels of total protein, albumin, total cholesterol, or triglyceride in male/female rats in any of the dose groups on day 28 compared with the control group. Since the liver is the major site of cholesterol synthesis, disposal, and degradation, the results demonstrated that the growing rats increased protein synthesis to an extent, and GYS had no effect on cholesterol metabolism in rats. The level of glucose (GLU) in both sexes decreased in the three dose groups compared with the control group (*p* < 0.05), suggesting that GYS had hypoglycemic activity. This may happen because of the increase clotting time and delay in centrifugation which results in glucose consumption by erythrocytes and low value of this parameter. In this study, apart from a significant reduction in the levels of serum glucose, no remarkable changes in the hematological and biochemical parameters were noted, representing that GYS was comparatively low toxic or nontoxic under the trial conditions.

From the necropsy assay, there were no lesions or pathological changes observed in the vital organs and reproduction organs of either sex rats in any of the dose groups on day 28 compared with the control group, indicating that GYS did not produce any treatment-associated toxicity or deterioration to the vital organs and reproduction organs function. In this study, the histopathological changes in the vital organs and reproduction organs of the rats in the three GYS-treated groups and the control group were also investigated, respectively. Compared with the control group, there were no toxicological or pathological changes in the organs after 28 consecutive days of GYS administration. The liver and kidney are sensitive organs, and their roles are affected by many factors, which include drugs, such as plant phytochemicals, eventually leading to renal dysfunction and liver toxicity. In this study, the histopathological assessment of the vital organs and reproductive organs in the rats which were given 30.0 g/kg BW GYS group shown that there were no treatment-related microscopic changes in the liver and kidneys. It also shows that the structures were normal and that there were no pathological lesions in the lungs, heart, spleens, thymus, uterus, ovaries, or testes. Based on these results, GYS seems to be low-toxic or nontoxic to the animals at all of the tested dosages, and did not cause any signs of toxicological or pathological changes. The present study is the first study to demonstrate that GYS, which is claimed to be acurative agent for uterine disorders in early postpartum cattle, is a medicinal plant extract with low-toxic or nontoxic properties.

## 5. Conclusions

In this study, the MTD of GYS was greater than 90.0 g/kg BW for the experimental mice in both sexes in the acute toxicity test. During the 28-day oral administration of GYS, the NOAEL of GYS was 30.0 g/kg BW in the subacute toxicity trial, which was 20 times greater than the dose that has already been used in clinical practice. No treatment-related deaths and no other clinical abnormalities were noted during the acute and subacute experimental periods. The blood hematology and serum biochemistry data revealed no differences between the treatment and control groups, and no toxic effects were detected in the vital organs and reproductive organs at any dose level by the histopathological examination. Therefore, the oral administration of GYS at a dose lower than 90.0 g/kg BW in one day or 30.0 g/kg BW per day for 28 days is safe for the mice or rats, which provided a basis for the clinical use of GYS and for determining a reasonable safe dose.

## Figures and Tables

**Figure 1 vetsci-10-00615-f001:**
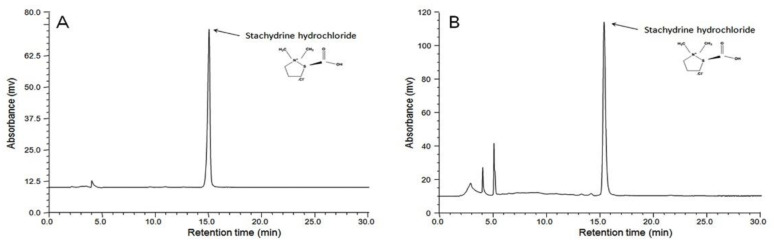
HPLC chromatogram for detecting stachydrine hydrochloride in GYS (Guixiong Yimu San). (**A**) Stachydrine hydrochloride control. (**B**) Sample of GYS.

**Figure 2 vetsci-10-00615-f002:**
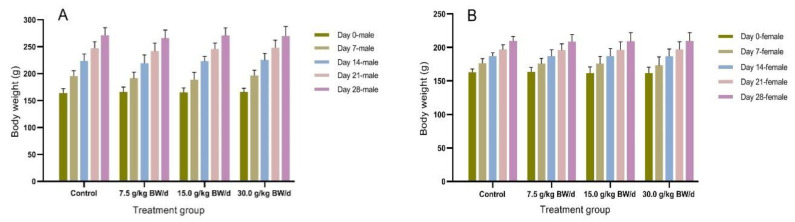
Body weight of control and treated male (**A**) and female (**B**) rats with 7.5, 15.0, and 30.0 g/kg BW of GYS (Guixiong Yimu San) for 28 days. There were no significant variations in treated groups relative to the control group (*p* > 0.05).

**Figure 3 vetsci-10-00615-f003:**
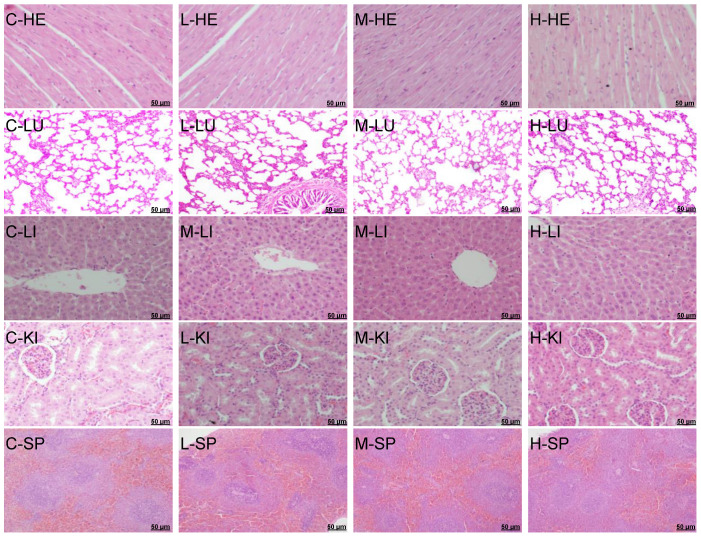
Representative photomicrographs of vital organs of the control (C) and GYS-treated rats (low dose, L; middle dose, M; high dose, H). Heart (HE, 400×); lung (LU, 200×); liver (LI, 400×); kidney (KI, 400×); spleen (SP, 100×). Scale bar = 50 µm. (H&E stain). No significant histopathological abnormality was noted in the vital organs of the treated and control rats.

**Figure 4 vetsci-10-00615-f004:**
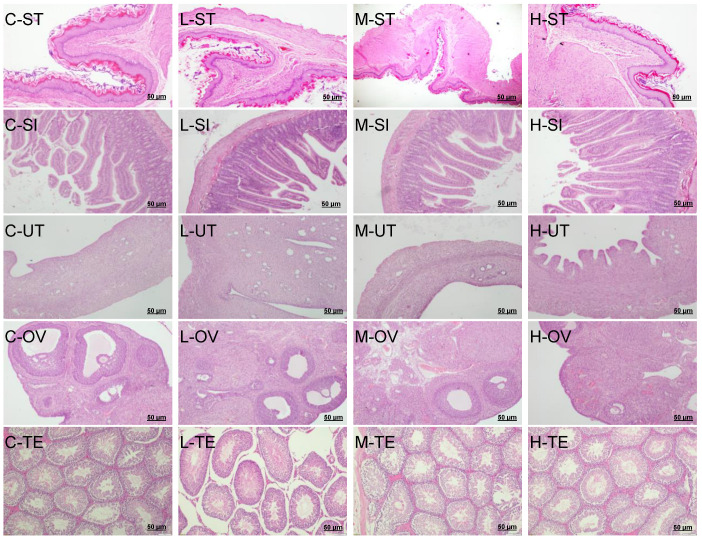
Representative photomicrographs of stomach, small intestine, and reproductive organs of the control (C) and GYS-treated rats (low dose, L; middle dose, M; high dose, H). Stomach (ST, 100×); small intestine (SI, 100×); uterus (UT, 100×); ovary (OV, 100×); testis (TE, 100×). Scale bar = 50 µm. (H&E stained). No remarkable histopathological abnormalities were noted in the control ortreated rats organs.

**Table 1 vetsci-10-00615-t001:** Effect on BW, BWG, and growth performance in rats after daily oral administration of GYS (n = 10).

Dose (g/kg BW/d)	Average Initial BW (g) ^1^	BWG (g/d/rat) ^2^				Average Final BW (g) ^3^	ADG (g) ^4^	ADFI (g) ^5^	FCR ^6^
		d 1–7	d 8–14	d 15–21	d 22–28				
**Male** ♂									
Control	163.94 ± 7.98	31.41 ± 5.51	28.06 ± 4.37	23.55 ± 6.73	24.00 ± 4.74	270.97 ± 14.32	3.82 ± 0.32	19.17 ± 1.68	5.06 ± 0.70
7.5	165.71 ± 9.38	31.64 ± 7.02	28.01 ± 4.73	22.33 ± 6.45	23.89 ± 4.21	265.58 ± 15.32	3.71 ± 0.48	18.76 ± 3.81	5.10 ± 0.98
15.0	165.29 ± 8.29	31.20 ± 8.12	28.35 ± 2.73	22.92 ± 4.02	21.79 ± 6.03	270.50 ± 14.42	3.73 ± 0.32	18.80 ± 0.87	5.09 ± 0.55
30.0	165.57 ± 7.31	30.65 ± 4.83	28.58 ± 6.41	22.82 ± 7.25	22.08 ± 5.89	269.90 ± 17.56	3.73 ± 0.45	18.52 ± 1.97	5.03 ± 0.76
**Female**♀									
Control	163.04 ± 4.89	13.33 ± 4.35	10.62 ± 5.28	9.70 ± 5.53	12.69 ± 4.05	209.88 ± 7.09	1.65 ± 0.30	13.29 ± 0.80	8.28 ± 1.48
7.5	163.29 ± 6.95	12.32 ± 4.18	11.19 ± 2.98	9.26 ± 3.35	12.28 ± 2.84	207.44 ± 11.06	1.61 ± 0.22	13.55 ± 0.42	8.59 ± 1.37
15.0	161.87 ± 8.95	13.71 ± 2.80	11.36 ± 2.93	9.23 ± 3.28	12.84 ± 2.94	209.01 ± 12.98	1.69 ± 0.23	13.69 ± 0.91	8.29 ± 1.38
30.0	161.67 ± 8.89	13.32 ± 3.86	11.74 ± 3.36	9.29 ± 1.74	12.72 ± 4.02	209.64 ± 12.26	1.71 ± 0.28	13.76 ± 0.71	8.25 ± 1.51

GYS, Guixiong Yimu San; BWG, body weight gain; ADG, average daily gain; ADFI, average daily feed intake. ^1–6^ Results were given as means ± SD (n = 10/sex/dose) and means with a different letter within a column are significantly different (*p* < 0.05). No statistical difference between the control and the three GYS-treated groups (*p* > 0.05). ^2, 4, 5, 6^ Growth performance of rats was assessed by measuring feed intake (FI), body weight gain (BWG), and feed conversion ratios (FCRs, FCR = ADFI/ADG). Body weight (BW) of rats per cage and feed consumption were recorded on day 1, 7, 14, 21, and day 28.

**Table 2 vetsci-10-00615-t002:** Effects of daily oral administration of GYS for 28 days on the hematological parameters of rats (n = 10).

Indexes	Reference Values	Groups and Treatments (g/kg BW/d)			
		Control	7.50 g/kg BW	15.0 g/kg BW	30.0 g/kg BW
**Male** ♂					
WBC (×10^9^/L) ^1^	2.3–13	5.01 ± 2.15	4.67 ± 0.67	4.69 ± 1.00	4.93 ± 0.61
LYM # (×10^9^/L) ^2^	1.9–11.0	4.32 ± 1.82	3.72 ± 0.31	3.91 ± 0.44	4.21 ± 0.58
MONO # (×10^9^/L) ^3^	0.00–0.54	0.25 ± 0.08	0.28 ± 0.03	0.27 ± 0.04	0.31 ± 0.09
GRAN # (×10^9^/L) ^4^	0.00–1.20	0.23 ± 0.06	0.20 ± 0.03	0.19 ± 0.10	0.20 ± 0.05
LYM (%) ^5^	51–91	87.14 ± 4.62	84.31 ± 5.81	83.66 ± 5.34	85.70 ± 6.22
MONO (%) ^6^	0–21	6.51 ± 0.99	6.84 ± 1.95	6.72 ± 1.16	6.93 ± 1.40
GRAN(%) ^7^	0–27	5.78 ± 1.38	5.65 ± 1.37	5.32 ± 1.15	5.39 ± 1.50
RBC (×10^12^/L) ^8^	5.0–8.5	7.64 ± 0.42	7.40 ± 0.54	7.47 ± 0.86	7.45 ± 0.58
HGB (g/dL) ^9^	11–16	15.49 ± 0.97	15.50 ± 1.14	15.54 ± 1.15	15.13 ± 1.23
HCT (%) ^10^	32–53	47.19 ± 1.92	46.57 ± 1.26	47.63 ± 1.53	46.76 ± 1.00
MCV (fL) ^11^	51–69	53.70 ± 2.60	53.67 ± 1.75	54.43 ± 2.81	53.49 ± 1.54
MCH (pg) ^12^	15–25	18.43 ± 0.62	18.71 ± 0.46	18.36 ± 0.48	18.73 ± 0.34
MCHC (g/L) ^13^	26–41	33.36 ± 0.56	33.39 ± 0.829	33.27 ± 0.92	32.33 ± 0.90
RDW (%) ^14^	10–18	17.22 ± 0.70	17.43 ± 1.37	17.41 ± 0.85	17.26 ± 1.40
PLT# (×10^9^/L) ^15^	538–1330	881.22 ± 64.54	873.57 ± 37.29	874.44 ± 38.88	878.25 ± 40.72
MPV (fL) ^16^	5.0–10.1	7.97 ± 0.55	8.01 ± 0.34	7.88 ± 0.47	8.13 ± 0.67
PDW (fL) ^17^	6.8–11.1	9.88 ± 0.28	9.50 ± 0.73	9.66 ± 1.36	9.80 ± 0.63
PCT (%) ^18^	0.4–0.8	0.64 ± 0.080	0.66 ± 0.04	0.63 ± 0.07	0.65 ± 0.04
**Female**♀					
WBC (×10^9^/L) ^1^	0.4–11	4.67 ± 0.90	4.47 ± 1.40	4.49 ± 1.19	4.50 ± 0.59
LYM # (×10^9^/L) ^2^	0.3–9.9	4.19 ± 0.47	3.96 ± 0.74	4.01 ± 0.62	4.03 ± 0.82
MONO # (×10^9^/L) ^3^	0.00–0.40	0.25 ± 0.04	0.27 ± 0.06	0.26 ± 0.06	0.25 ± 0.04
GRAN # (×10^9^/L) ^4^	0.00–0.76	0.22 ± 0.03	0.20 ± 0.05	0.20 ± 0.01	0.21 ± 0.06
LYM (%) ^5^	54–88	89.32 ± 2.91	87.10 ± 5.48	86.25 ± 5.30	85.78 ± 5.48
MONO (%) ^6^	0–21	6.41 ± 1.02	6.77 ± 0.85	6.75 ± 1.21	6.88 ± 1.35
GRAN (%) ^7^	0–27	4.98 ± 0.84	4.64 ± 0.80	4.71 ± 1.78	4.68 ± 1.28
RBC (×10^12^/L) ^8^	6.0–8.2	7.99 ± 1.26	7.57 ± 0.65	7.85 ± 1.43	7.65 ± 1.40
HGB (g/dL) ^9^	12–16	14.61 ± 2.22	14.53 ± 1.52	14.43 ± 1.34	14.51 ± 0.64
HCT (%) ^10^	32–53	43.32 ± 7.19	43.70 ± 2.58	44.76 ± 3.27	42.73 ± 2.14
MCV (fL) ^11^	51–65	57.14 ± 1.10	57.24 ± 1.14	57.45 ± 1.25	58.09 ± 1.75
MCH (pg) ^12^	17–22	18.75 ± 0.55	18.50 ± 0.44	18.56 ± 0.87	18.74 ± 0.53
MCHC (g/L) ^13^	32–36	33.83 ± 0.82	33.67 ± 0.85	33.27 ± 1.31	33.57 ± 1.34
RDW (%) ^14^	10–18	16.80 ± 0.89	16.54 ± 1.22	16.63 ± 1.46	16.98 ± 0.90
PLT (×10^9^/L) ^15^	600–1290	785.30 ± 102.65	768.43 ± 73.66	771.75 ± 118.74	777.11 ± 127.47
MPV (fL) ^16^	5.0–8.7	7.98 ± 0.58	8.16 ± 0.33	8.11 ± 0.49	8.13 ± 0.67
PDW (fL) ^17^	6.8–11.1	9.77 ± 0.74	9.69 ± 0.64	9.70 ± 1.33	9.66 ± 1.17
PCT (%) ^18^	0.4–0.8	0.58 ± 0.20	0.61 ± 0.06	0.56 ± 0.24	0.63 ± 0.12

WBC: white blood cell; LYM: lymphocyte differentiation; MONO: monocyte differentiation; GRAN: granulocyte differentiation; LYM#: lymphocyte counts; MONO#: monocyte counts; GRAN#: granulocyte counts; RBC: red blood cell; HGB: hemoglobin concentration; HCT: hematocrit; MCV: mean corpuscular volume; MCH: mean corpuscular hemoglobin; MCHC: mean corpuscular hemoglobin concentration; RDW: red blood cell distribution width; PLT: platelets; MPV: mean platelet volume; PDW: platelet distribution width; PCT: platelet cubic measure distribution width. ^1–18^ Data were presented as means ± SD (n = 10/sex/dose, after 28-day administration), the mean difference is significant at the 0.05 level. No statistical difference between the control and the three GYS-treated groups (*p* > 0.05).

**Table 3 vetsci-10-00615-t003:** Effects of daily oral administration of GYS for 28 days on the biochemical parameters of rats (n = 10).

Indexes	Reference Values	Groups and treatments (g/kg BW/d)			
		Control	7.50 g/kg BW	15.0 g/kg BW	30.0 g/kg BW
**Male** ♂					
AST (U/L) ^1^	75–278	262.36 ± 26.96	263.26 ± 7.61	261.75 ± 41.39	259.73 ± 20.61
ALT (U/L) ^2^	19–146	23.47 ± 2.08	22.91 ± 1.95	22.13 ± 1.86	23.12 ± 5.13
ALP (U/L) ^3^	41–138	119.71 ± 15.92	120.91 ± 8.99	118.13 ± 16.26	116.11 ± 11.91
TBL (µmol/L) ^4^	0.3–0.7	0.82 ± 0.29	0.81 ± 0.07	0.83 ± 0.18	0.80 ± 0.44
BUN (mg/dL) ^5^	4–10	7.30 ± 0.68	7.53 ± 0.42	7.14 ± 0.99	7.38 ± 1.01
CRE (µmol/L) ^6^	34–63	45.12 ± 4.05	43.94 ± 5.26	43.97 ± 3.54	45.05 ± 6.65
TC (mg/dL) ^7^	18.2–41.6	22.70 ± 5.60	22.44 ± 2.02	23.11 ± 1.69	21.66 ± 4.10
TG (mmol/L) ^8^	0.4–1.3	0.68 ± 0.15	0.70 ± 0.14	0.68 ± 0.10	0.67 ± 0.10
GLU (mmol/L) ^9^	4–8	5.71 ± 0.99 ^a^	5.34 ± 0.68 ^b^	5.29 ± 0.64 ^b^	5.17 ± 0.71 ^b^
TP (g/L) ^10^	47–59	52.14 ± 2.50	50.91 ± 2.81	51.79 ± 3.17	51.58 ± 3.35
ALB (g/L) ^11^	18–41	26.62 ± 2.27	25.57 ± 1.35	26.96 ± 1.44	25.06 ± 1.39
**Female** ♀					
AST (U/L) ^1^	57–258	261.80 ± 43.84	264.90 ± 47.02	261.04 ± 32.15	266.76 ± 54.47
ALT (U/L) ^2^	20–85	25.15 ± 3.12	25.81 ± 2.13	24.97 ± 5.05	24.14 ± 4.07
ALP (U/L) ^3^	32–75	54.61 ± 4.19	53.20 ± 5.67	54.25 ± 11.98	53.89 ± 17.23
TBL (µmol/L) ^4^	0.4–0.8	0.58 ± 0.16	0.56 ± 0.20	0.51 ± 0.10	0.53 ± 0.16
BUN (mg/dL) ^5^	5–12	7.90 ± 0.76	7.86 ± 1.12	7.91 ± 0.64	7.90 ± 0.41
CRE (µmol/L) ^6^	36–68	46.22 ± 9.24	45.34 ± 5.03	44.58 ± 8.54	46.41 ± 5.15
TC (mg/dL) ^7^	12.6–46.8	23.44 ± 3.34	24.17 ± 4.49	24.75 ± 3.21	23.80 ± 2.05
TG (mmol/L) ^8^	0.2–1.3	0.63 ± 0.09	0.67 ± 0.13	0.66 ± 0.04	0.62 ± 0.06
GLU (mmol/L) ^9^	4–8	5.21 ± 0.69 ^a^	4.87 ± 0.37 ^b^	5.01 ± 0.97 ^b^	4.98 ± 0.48 ^b^
TP (g/L) ^10^	50–65	52.08 ± 2.96	51.00 ± 3.01	53.49 ± 1.71	51.17 ± 4.97
ALB (g/L) ^11^	20–44	27.93 ± 2.22	27.15 ± 1.09	28.10 ± 0.83	26.65 ± 1.33

AST: aspartate aminotransferase; ALT: alanine aminotransferase; ALP: alkaline phosphatase; TBL: total bilirubin; BUN: blood urea nitrogen; CRE: creatinine; TC: total cholesterol; TG: triglyceride; GLU: glucose; TP: total protein; ALB: albumin. ^1–11^ Data were presented as means ± SD (n = 10/sex/dose, after 28-day administration), the mean difference is significant at the 0.05 level. ^a b^ Similar superscript showing no statistically remarkable differences between the groups (*p* > 0.05) and vice versa.

**Table 4 vetsci-10-00615-t004:** Effect of GYS on relative organ weights (ROW) of rats after 28 days of oral administration (n = 10).

Indexes	Groups and Treatments (g/kg BW/d)			
	Control	7.50 g/kg BW	15.0 g/kg BW	30.0 g/kg BW
**Male** ♂				
Heart ^1^	0.35 ± 0.04	0.34 ± 0.02	0.34 ± 0.05	0.35 ± 0.02
Lung ^2^	0.52 ± 0.03	0.53 ± 0.08	0.53 ± 0.08	0.53 ± 0.06
Liver ^3^	3.07 ± 0.40	3.07 ± 0.32	3.06 ± 0.29	3.29 ± 0.30
Kidney ^4^	0.79 ± 0.08	0.80 ± 0.03	0.81 ± 0.09	0.81 ± 0.07
Spleen ^5^	0.23 ± 0.04	0.21 ± 0.03	0.21 ± 0.02	0.22 ± 0.04
Thymus ^6^	0.17 ± 0.03	0.16 ± 0.03	0.16 ± 0.05	0.17 ± 0.03
Uterus ^7^	0.20 ± 0.06	0.20 ± 0.04	0.21 ± 0.05	0.21 ± 0.05
**Female** ♀				
Heart ^1^	0.33 ± 0.06	0.33 ± 0.01	0.36 ± 0.03	0.34 ± 0.02
Lung ^2^	0.57 ± 0.08	0.59 ± 0.09	0.54 ± 0.06	0.58 ± 0.08
Liver ^3^	2.79 ± 0.40	2.78 ± 0.15	2.96 ± 0.23	2.95 ± 0.35
Kidney ^4^	0.78 ± 0.10	0.77 ± 0.05	0.78 ± 0.06	0.83 ± 0.07
Spleen ^5^	0.23 ± 0.04	0.21 ± 0.01	0.23 ± 0.03	0.24 ± 0.06
Thymus ^6^	0.21 ± 0.07	0.20 ± 0.04	0.19 ± 0.05	0.17 ± 0.03
Testis ^7^	1.22 ± 0.04	1.24 ± 0.04	1.23 ± 0.04	1.20 ± 0.12

^1–7^ Data were presented as means ± SD (n = 10/sex/dose, per group for male and female rats, after 28-day administration), the mean difference is significant at the 0.05 level. No statistically significant differences were found (*p* > 0.05).

## Data Availability

The datasets generated for this study are available on request to the corresponding author.

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
