# Peer review of "Acute and 28-Day Repeated-Dose Oral Toxicity of the Herbal Formula Guixiong Yimu San in Mice and Sprague–Dawley Rats"

_vetsci, 2023, doi:10.3390/vetsci10100615_

Round 1
Reviewer 1 Report
This is an informative manuscript with beautiful data, which helps people gain knowledge about the toxicity and provides a guidance for the dosage of GYS in clinical practice.
I have a few suggestions about the manuscript:
(1) Introduction: Authors have better to provide some background info about the dosage of GYS in clinical practice, and make a connection to corresponding experimental design. Can authors also give more info about each component (functions?) of the GYS and why people only focus on the active compound of Leonuri herba.
(2) HPLC analysis: Can authors provide a gold standard to evaluate the measured value of stachydrine hydrochlorid in GYS and determine its quality?
(3) Acute toxicity test of GYS in mice: Given the high turnover rate of the cells in some organs like intestine, 14-day waiting period are long enough for the regeneration of their structures, which may lead to "no pathological changes".
(4) Repeated-dose 28-day oral toxicity study of GYS in rats: Can authors give the reason why they change the model to rat in chronic study? for gavage safety purpose?
I also add some comments and point out some minor errors in the PDF file attached.

I can understand the subject matter of the manuscript, but some grammar errors cannot be ignored.
Author Response
Thank you very much for taking the time to review this manuscript. Please find the detailed responses below and the corresponding revisions/corrections which are highlighted in red color in the re-submitted files.
3. Point-by-point response to Comments and Suggestions for Authors |
Comment 1: Replaced with a comma |
Response 1: Replaced as per suggestion |
Comment 2: 30, 15, 7.5? |
Response 2: I agree and the mistake is corrected |
Comment 3: Grammar check |
Response 3: Checked and corrected |
Comment 4: Citations |
Response 4: Added |
Comment 5: Can you provide the efficacious dosage for cow as a reference? |
Response 5: The relevant information is written under discussion section. |
Comment 6: 14-day waiting period are long enough for the regeneration |
Response 6: For the acute toxicity test (MTD method), I refer to Wan, L. et al. (reference 31), the duration of their trial was 14 days. |
Comment 7: Can you provide a typical treatment scheme of the GYS in retained placenta? Such as dosage / time for acute case and dosage / time for chronic case? |
Response 7: The relevant information is written under discussion section. I think this comment not belong to this section as the experiment is about the oral toxicity of GYS in rats, not about the retained placenta. Moreover, I will appreciate if you guide me further for more clarity. |
Comment 8: Can you add a justification about the reason why using mice in acute study while using rats in chronic study. |
Response 8: According to the requirements of the Chinese Veterinary Pharmacopoeia (2015 edition, Ministry of Agriculture) "Technical Guidelines for long-term Toxicity Research of Natural drugs for veterinary use" and the declaration requirements of Class II innovative veterinary drugs, acute oral toxicity test can be carried out in mice; 28-day repeated-dose oral toxicity test can be performed on rats. |
Comment 9: Percentage? do you have a gold standard to compare? |
Response 9: This is the amount of stachydrine hydrochloride per gram of GYS, not the percentage concentration. Leonuri herba (Yi-Mu-Cao, YMC; 46.15%, w/w; JUNYAO, Monarch drug), is the key component of Guixiong Yimu San (GYS). In accordance with the composition of traditional Chinese medicine formula, the preferred principle of Monarch drug (JUNYAO), and the instruction of the Committee for the Veterinary Pharmacopoeia of China (2015 version), stachydrine hydrochloride, biologically active chemical compound of Leonuriherba (the key component of herbal formula)—was indicated to be a biomarker of GYS for qualitative identification and quantitative analysis. The content of stachydrine hydrochloride is an important index component to measure the quality of GYS. The quantitative study of stachydrine hydrochloride by HPLC method can effectively control the quality of GYS. The HPLC method for the determination of stachydrine hydrochloride in Leonuri herba (Yi-Mu-Cao) based on China Veterinary Pharmacopoeia (2015 edition). |
Comment 10: Any discussion for this phenomena? |
Response 10: Discussed in discussion section and highlighted in red color. |
Comment 11: Citation |
Response 11: Added |
Comment 12: Gold standard to compare with? |
Response 12: Already justified earlier. |
Comment 13: Better insert some dosage info in the introduction and build the connection about the experimental design. |
Response 13: Added in the introduction section and highlighted in red color. |
Comment 14: Introduction: Authors have better to provide some background info about the dosage of GYS in clinical practice, and make a connection to corresponding experimental design. Can authors also give more info about each component (functions?) of the GYS and why people only focus on the active compound of Leonuri herba. |
Response 14: Yes, we have added in introduction section in red color. A traditional herbal formula generally contains different quantities of several herbs with different roles (Monarch, Minister, Adjuvant and Guide). In our herbal powder, the large dose of Leonuriherba (Yi-Mu-Cao, YMC; Leonurus ja ponicus Houtt.) can modulate uterine contractions and improve uterine hemorheology and microcirculation against the principal symptom of cows affected with retained placenta. Angelicae sinensis radix (Dang-Gui, DG; Angelica sinensis (Oliv.) Diels), Chuanxiong rhizoma (Chuan-Xiong, CX; Ligusticum chuanxiong Hort.), and Carthamiflos (Hong-Hua, HH; Carthamus tinctorius L.), were used for activating blood circulation to remove blood stasis, which assists the Leonurus artemisia (Laur.). Cyperirhizoma (Xiang-Fu, XF; Cyperusrotundus L.) in regulating uterine contraction and improving the uterine condition. Glycyrrhizaeradix et rhizoma (Gan-Cao, GC; Glycyrrhiza uralensis Fisch), a tonifying herbal medicine, harmonizes the herbs and minimizes any potential side effects. All of these herbs work together as a formula to vitalize the blood, transform stasis, contract the uterus, warm the channels and alleviate pain to resolve postpartum blood stasis syndrome. |

Reviewer 2 Report
In this article, a phytocompound-based veterinary drug is analyzed for the treatment of retained placenta, and its toxicity is analyzed.
This manuscript is technically correct, providing important information for a comune condition in the veterinary field, Guixiong Yimu San (GYS) being a good treatment option if efficacy and reduced toxicity are demonstrated. As long as the medical efficacy is mentioned and demonstrated in other studies, this manuscript insists on its toxicity.
I think the article can be published after some fine corrections (if there were line numbers, the correction would have been easier):
(In Abstract)
At the given dose, there were no mortality or signs of toxicity throughout the 28-day subacute toxicity study. à There were no mortality or signs of toxicity at the given dose throughout the 28-day subacute toxicity study
(Introduction)
containied à contained
therfore à therefore
(Materials and metods)
Stock solution of the reference standards was prepared at suitable concentrations (0.0375 mg/mL,
à The stock solution....
The conclusions can be improved and detailed.
The correlation of the biochemical and histopathological results obtained for the toxicity aspects supports the scientific quality of the article, it can be published after some corrections of technical editing according to the rigors of the journal.
-
Author Response
Thank you very much for taking the time to review this manuscript. Please find the detailed responses below and the corresponding revisions/corrections which are highlighted in green color in the re-submitted files.
Comment 1: In abstract, at the given dose, there were no mortality or signs of toxicity throughout the 28-day subacute toxicity study. |
Response 1: Yes, no treatment-related deaths or behavioral changes occurred during the experimental period. |
Comment 2: In materials and methods, stock solution of the reference standards was prepared at suitable concentrations (0.0375 mg/mL, w/v) in methanol and stored at 4°C until analysis.) |
Response 2: Yes, the concentration of the reference solution of stachylonine hydrochloride was 0.0375 mg /mL. |
Comment 3: The conclusions can be improved and detailed. |
Response 3: Thanks, checked and corrected |
Comment 4: The correlation of the biochemical and histopathological results obtained for the toxicity aspects supports the scientific quality of the article, it can be published after some corrections of technical editing according to the rigors of the journal. |
Response 4: Thank you very much for reviewing this article and giving your suggestions. |

Reviewer 3 Report
1. What device was used to administer to the rats?.
2. What parameters or variables were considered to carry out the behavioural recording?.
3. Euthanized.
4. Was any staining used to carry out the histopathological analysis?.
5. Why did this group receive distilled water and not physiological saline?.
6. Did these animals undergo any fasting prior to blood biometry and blood chemistry?.
7. Does this equipment the biological reference intervals for the sex and strain of rat and mouse used for this project? These intervals are very important, as they usually use human biological reference intervals and make comparisons with samples from experimental subjects. If not, this situation must be addressed because it will have a direct impact on the results obtained.
8. It was a long time to wait for clot formation, a rat's blood takes between 10 and 15 minutes maximum to clot, and as described in the summary and their results, glucose values were decreased in all three groups of males and females.
9. For the bilirubin determination, what care was taken with the sample?.
10. In table 2. A column should be added with the biological reference intervals for each of the parameters that make up the blood biometry and that correspond to the Sprague-Dawley strain since relying on control rats is no guarantee that these animals are healthy, and therefore their intervals vary. Furthermore, when carrying out this type of study, it must be considered that the parameters that will serve as the biological reference interval are from the strain of animals we are using.
11. In table 3. A column should be added with the biological reference intervals for each of the biochemical parameters that correspond to the Sprague-Dawley strain since relying on control rats is no guarantee that these animals are healthy, and therefore their intervals vary. Furthermore, when carrying out this type of study, it must be considered that the parameters that will serve as the biological reference interval are from the strain of animals we are using.
12. In discussion: As I mentioned previously, because there was a delay in centrifuging the blood samples and because they waited 2 hours to clot, therefore the erythrocytes consumed the glucose and this had an impact on the low values of this parameter.

Author Response
Thank you very much for taking the time to review this manuscript. Please find the detailed responses below and the corresponding revisions/corrections highlighted in blue color in the re-submitted files.
Point-by-point response to Comments and Suggestions for Authors |
Comments 1: What device was used to administer to the rats? |
Response 1: Oral gavage needle |
Comment 2: What parameters or variables were considered to carry out the behavioral recording? |
Response 2: The behavioral parameters were noticed such as tremors, convulsions, diarrhea, lethargy, and coma. |
Comment 3: Euthanized |
Response 3: Corrected as per suggestion. All mice that survived were euthanized by anesthetic overdose of sodium pentobarbital (150 mg/kg BW intraperitoneal, i.p.). |
Comment 4: Was any staining used to carry out the histopathological analysis? |
Response 4: No, during the oral acute toxicity test, none of the mice died; there were no significant changes in the general behavior, fur color, mucosa color, and posture; there were no lesions or pathological changes were observed in the vital organs and reproduction organs. |
Comment 5: Why did this group receive distilled water and not physiological saline? |
Response 5: We are sorry for writing mistake, the control group received physiological saline (0.9% NS). |
Comment 6: What parameters or variables were considered to carry out the behavioral recording? |
Response 6: e.g., tremors, convulsions, diarrhea, lethargy, and coma. |
Comment 7: euthanized |
Response 7: Yes, corrected as per suggestion. |
Comment 8: Did these animals undergo any fasting prior to blood biometry and blood chemistry? |
Response 8: Yes, it’s written in methodology and highlighted in blue color. |
Comment 9: Mention the times exactly. |
Response 9: Mentioned. The exact time was 16 hours. |
Comment 10: Does this equipment the biological reference intervals for the sex and strain of rat and mouse used for this project? These intervals are very important, as they usually use human biological reference intervals and make comparisons with samples from experimental subjects. If not, this situation must be addressed because it will have a direct impact on the results obtained. |
Response 10: Yeah, the suitable statement also added in the text for clarification. |
Comment 11: It was a long time to wait for clot formation, a rat's blood takes between 10 and 15 minutes maximum to clot, and as described in the summary and their results, glucose values were decreased in all three groups of males and females. |
Response 11: We agree with you, but we were thinking that the blood will take more time to clot because GYS also act as anticoagulant and may enhance the blood clotting time. That’s why we have given more time to clot. |
Comment 12: For the bilirubin determination, what care was taken with the sample? |
Response 12: The sample was protected from light by using aluminum foil. Also analyze the sample within 2 hours, if possible or store at low temperature (‒80℃) if delay in processing. |
Comment 13: Does this equipment the biological reference intervals for the sex and strain of rat and mouse used for this project? These intervals are very important, as they usually use human biological reference intervals and make comparisons with samples from experimental subjects. If not, this situation must be addressed because it will have a direct impact on the results obtained. |
Response 13: Yeah, the suitable statement also added in the text for clarification. |
Comment 14: euthanized |
Response 14: Corrected as per suggestion. |
Comment 15: euthanized |
Response 15: Corrected as per suggestion. |
Comment 16: A column should be added with the biological reference intervals for each of the biochemical parameters that correspond to the Sprague-Dawley strain since relying on control rats is no guarantee that these animals are healthy, and therefore their intervals vary. Furthermore, when carrying out this type of study, it must be considered that the parameters that will serve as the biological reference interval are from the strain of animals we are using. |
Response 16: Yes, thanks for your suggestion, we have added a column with reference range for each of parameters. |
Comment 17: A column should be added with the biological reference intervals for each of the biochemical parameters that correspond to the Sprague-Dawley strain since relying on control rats is no guarantee that these animals are healthy, and therefore their intervals vary. Furthermore, when carrying out this type of study, it must be considered that the parameters that will serve as the biological reference interval are from the strain of animals we are using. |
Response 17: Yes, we have added a column with reference range for each of parameters. |
Comment 18: As I mentioned previously, because there was a delay in centrifuging the blood samples and because they waited 2 hours to clot, therefore the erythrocytes consumed the glucose, and this had an impact on the low values of this parameter. |
Response 18: Yes, you are right. We have added this justifying statement about this parameter. |
